# Proactively location-based suppression elicited by statistical learning

**Siyang Kong[1], Xinyu Li[1], Benchi Wang**[ORCID][2,3,4,5]\*, **Jan Theeuwes**[1,6,7]

**1** Department of Psychology, Zhejiang Normal University, Jinhua, China, **2** Institute for Brain Research and Rehabilitation, South China Normal University, Guangzhou, China, **3** Center for Studies of Psychological Application, South China Normal University, Guangzhou, China, **4** Guangdong Key Laboratory of Mental Health and Cognitive Science, South China Normal University, Guangzhou, China, **5** Key Laboratory of Brain, Cognition and Education Sciences (South China Normal University), Ministry of Education, Guangzhou, China, **6** Department of Experimental and Applied Psychology, Vrije Universiteit Amsterdam, Amsterdam, The Netherlands, **7** Institute Brain and Behavior Amsterdam, Amsterdam, The Netherlands

\* wangbenchi.swift@gmail.com

**Data Availability Statement:** Data and procedure can be accessed through https://github.com/wangbenchi/Search_probe.

**Funding:** yes, XL, No. LY18C090007, the Natural Science Foundation of Zhejiang Province, http://www.zjzwfw.gov.cn; BW, No.2019A1515110581,

## Abstract

Recently, Wang and Theeuwes used the additional singleton task and showed that attentional capture was reduced for the location that was likely to contain a distractor [1]. It is argued that due to statistical learning, the location that was likely to contain a distractor was suppressed relative to all other locations. The current study replicated these findings and by adding a search-probe condition, we were able to determine the initial distribution of attentional resources across the visual field. Consistent with a space-based resource allocation ("biased competition") model, it was shown that the representation of a probe presented at the location that was likely to contain a distractor was suppressed relative to other locations. Critically, the suppression of this location resulted in more attention being allocated to the target location relative to a condition in which the distractor was not suppressed. This suggests that less capture by the distractor results in more attention being allocated to the target. The results are consistent with the view that the location that is likely to contain a distractor is suppressed before display onset, modulating the first feed-forward sweep of information input into the spatial priority map.

## Introduction

It is important to be able to attend to events that are relevant to us and ignore information that may distract us. Typically, salient objects in the environment have the ability to automatically grab our attention and disrupt our ongoing tasks [2, 3]. The extent to which we are able to avoid such distraction from salient events has been a central question for decades. Traditionally, it was assumed that the competition between top-down, goal-directed signals and bottom-up, salience-based signals determined the selection priority in the visual field [for reviews see 2, 4]. Recently, it was pointed out that a third category labeled "selection history" plays a larger role than previously assumed [5]. It was argued that the repeated exposure to stimuli creates (often implicitly) learned selection biases, shaped by the repeated associations of value, emotion valence, or other statistical regularities [3, 6, 7]. These effects cannot be explained by

the Natural Science Foundation of Guangdong Province. The funders had no role in study design, data collection and analysis, decision to publish, or preparation of the manuscript.

**Competing interests:** The authors have declared that no competing interests exist.

top-down nor by bottom-up factors. As such, these learning processes provide competitive advantages for certain spatial locations and/or visual features by means of altering their priority for attentional selection [1, 8–15].

Extracting regularities from the environment in service of automatic behavior is one of the most fundamental abilities of any living organism and is often referred to as statistical learning (SL). Statistical learning has been a subject of investigation in many domains, particularly in language acquisition, object recognition, attention, scene perception, visual search, conditioning and motor learning [16–24].

Previous research on statistical learning and attention has shown that observers can learn to prioritize locations that are likely to contain a target. For example, learning contextual regularities biases attentional selection such that searching for a target is facilitated when it appears in a visual lay-out that was previously searched relative to visual lay-outs that were never seen before; research known under the term of "*contextual cueing*" [25–28]. Typically, in these studies, participants searched for a 'T' target among 'L' distractors in sparsely scattered configurations. Half of the display configurations were repeated across blocks while others were only seen once. The classic result was that participants were faster in finding targets when they appeared in repeated configurations than in configurations that they had not seen before, suggesting that participants have learned the association between the spatial configuration and the target location. These studies show that observers can learn the association between display configurations and the target location, consistent with findings that have shown that observers are faster to detect targets appearing in probable locations than improbable locations [29, 30]. Consistent with this notion are studies that have shown that observers are faster to respond to targets that appear at more probable locations than in all other locations [31–33].

Recently, however, it has been shown that lingering biases due to statistical learning history play an important role in avoiding and/or reducing distraction. In a series of experiments, Wang & Theeuwes employed a variant of the additional singleton task and showed that through statistical learning, attentional capture by the salient distractors could be significantly reduced [1, 14, 15]. Specifically, in these experiments, participants searched for a salient shape singleton (i.e., a diamond between circles or a circle between diamonds) while they ignored a salient colored distractor singleton. Critically and unknown to the participants, the presentation of the salient distractors was biased such that it was more likely to appear at one specific location (high-probability location) than at all other locations (low-probability location) in the visual field. The results indicated that there was less capture by the salient distractor when it appeared at this high-probability location than low-probability locations, suggesting that capture by salient distractor was attenuated. Moreover, when the target happened to be presented at the high-probability location, its selection was less efficient (in terms of RT and accuracy). In all studies, there was also a spatial gradient from the high-probability location as the attentional capture effect scaled with the distance from this location, and observers were basically unaware of the regularities.

These findings have led to the conclusion that the exposure to regularities regarding a distractor induces spatially selective suppression [12, 34–36]. Critically, this suppression is not found when participants actively try to suppress such a location in a top-down fashion [14, 37]. Importantly, a recent EEG study employing the same paradigm as in Wang and Theeuwes [1] showed that ~1200 ms before display onset, there was increased alpha power contralateral to the high-probability location relative to its ipsilateral location [38]. This type of alpha-band oscillations has been associated with neural inhibition serving as an attentional gating mechanism [39]. These neural signatures suggest that, well before the display is presented, the location that is likely to contain a distractor is suppressed. Because the location is suppressed before display onset, this type of suppression is referred to as "proactive suppression", suggesting that on the spatial priority map this location competes less for spatial attention than all

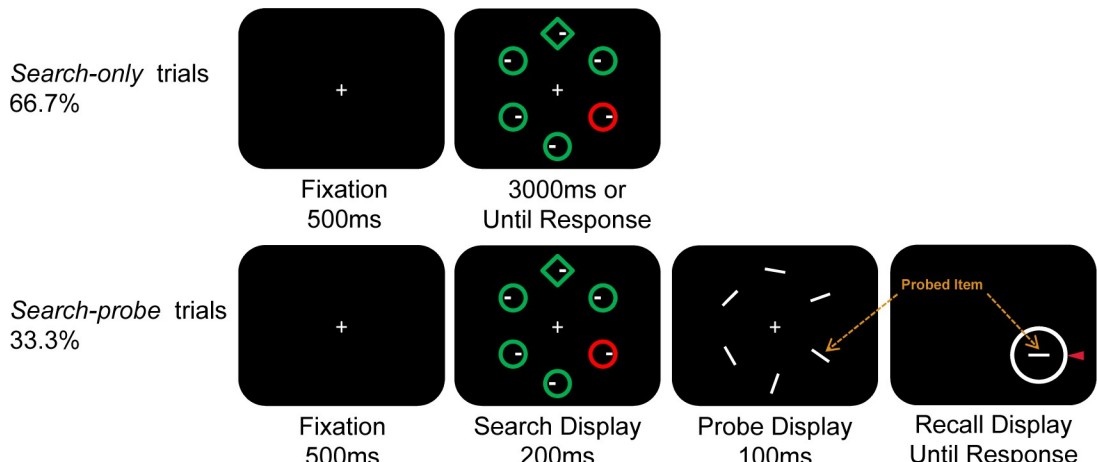

**Fig 1. Experimental procedure.** Search-only trials, in which participants were required to search for a shape singleton and to indicate the position (i.e., left or right) of the white dot inside. Search-probe trials, in which the search display was present for a short period (200 ms), then participants were required to memorize six orientations and to recall one of them by rotating the response wheel as accurate as possible. This figure was used for illustration purpose only, for details see the text.

other locations in the visual field [3]. Proactive suppression can be contrasted with retroactive suppression, which is the type of suppression that occurs only after attention has been directed to a location, disengaged and subsequently suppressed [13].

In the current study, we used a probe task to investigate the nature of this proactive suppression effect further. Participants performed a variant of Wang and Theeuwes [1] task in which the distractor singleton was presented more often in one location (high-probability location) than in all other locations (low-probability location). This task was performed on the majority of trials (66.7%). In the remaining trials, the search display was presented briefly (200 ms) immediately followed by a probe display, in which six orientation bars were presented [For an illustation see Fig 1; and see a similar probe task design in 40, 41]. Following the probe display, a bar appeared at one of the 6 locations, and participants had to adjust the bar such that its orientation would match the orientation of the element in the corresponding probe display. The distribution of response errors (i.e., response orientation value minus the correct orientation value) was characterized by fitting a standard mixture model [42] allowing us quantify independently measures of guess rate and standard deviation. This will allow us to examine whether the observed suppression effect was due to lower chance to encode the items (reflected by guess rate), or whether inhibition occurred after processing the items (reflected by standard deviation).

According to the biased competition theory of attention, objects in the visual field compete for cortical representation in a mutually inhibitory network [43]. Directing attention to one object comes at a cost of less attention for other objects. Bahcall and Kowler offered a similar space-based, resource allocation account, arguing that at the attended location, processing strength is increased by borrowing resources from other regions in the visual field [44]. Important for the current study, according to a biased competition resource allocation model of attention, proactive suppression of a particular location should result in more resources being available for target processing [38].

The probe task allows us to examine this distribution of attention across the visual field immediately following the display onset during the first 200 ms [41, 45]. If the location that is likely to contain a distractor is already suppressed before display onset (i.e., proactively), then one expects that a probe presented at that location is suppressed, resulting in a poor probe representation and consequently more probe response errors. Specifically, this poorer

representation should result in a lower chance to encode the items resulting in a higher guess rate; if the suppression is retroactive and occurring later in time we expect effects on the standard deviation. At the same time, according to a biased competition (resource) account, this proactive suppression should allow more resources being available for target processing, resulting in a better probe representation and fewer probe response errors, and lower guess rate. The reverse conjecture should also hold: if a distractor is presented at a low-probability location, it is basically not inhibited leading to stronger attentional capture than a distractor is presented at a high-probability location.

Attentional capture implies that attention is directed to this location, resulting in a better representation of the probe presented at that location (i.e., fewer probe response errors), while at the same time less resources should be available for target processing leading to more probe response errors [46]. Previous studies [1, 8, 14–15] only showed that capture was reduced when the distractor was presented the high-probability location relative to the low-probability location. The current study goes much beyond these findings and provide insights on how statistical learning impacts the distribution of attention across the display. Overall, we claim that the probe task adopted in the present study could provide a window of how, due to statistical learning, the weights within the spatial priority map are changed.

## Method

### Participants

Sixteen undergraduates (1 man and 15 women: with a mean age of 18.9 ± 1.0 years old) were recruited from Zhejiang Normal University in China. All participants provided written informed consent, and reported normal color vision and normal or corrected-to-normal visual acuity. Sample size was predetermined based on the significant difference between the high-probability location and low-probability location in Wang and Theeuwes [1], with an effect size of 1.83. With 16 subjects and alpha = .001, power for the critical effect should be > 0.99. The study was approved by both the Ethical Review Committee of the Vrije Universiteit Amsterdam and the Ethical Review Committee of Zhejiang Normal University.

### Apparatus and stimuli

Stimulus presentation and response registration were controlled by custom scripts written in Python 2.7. In a dimly lit laboratory, participants held their chins on a chin rest located 63 cm away from the liquid crystal display (LCD) color monitor. The primary search display contained one circle with a radius of 0.7˚ and five diamonds (subtended by 1.6˚ × 1.6˚) colored in red or green, or vice versa (see Fig 1 for an example). Each display-element was centered 2.0˚ from the fixation (a white cross, 0.5˚ × 0.5˚), containing a 0.2˚ white dot located 0.2˚ from either the left or the right edge of the element.

In search-probe trials, six white lines (subtending 0.1˚ × 1˚) with different orientations (randomly selected from seven orientations: 10˚-160˚, in 25˚ steps) were presented at the same locations as the display-elements of the search array (see Fig 1 bottom panel). A continuous response wheel (subtending 0.5˚ wide, 4.5˚ radius) presented at the center of the to-be-recalled item and a red pointer were used to collect participants' response.

### Procedure and design

On each trial, a fixation cross was presented for 500 ms, followed by a primary search display which consisted of five items in the same shape and a shape singleton (i.e., a circle among five diamonds, or vice versa). Participants were asked to keep fixation at the cross throughout the

trial. In search-only trials (66.7% of the trials), the search array was presented for 3000 ms or until participants responded. Participants were required to search for one shape singleton and indicate whether the line segment was on the left or right side of the target by pressing the left or right key on the keyboard using left hand as fast as possible, respectively. Responses were speeded and feedback, "You did not respond, please focus on the task" or "you responded incorrectly, please focus on the task" was given when participants did not respond or responded incorrectly, respectively.

In search-probe trials (33.3% of the trials), the search array appeared for 200 ms, followed by 100 ms probe display containing six orientation bars. Participants were required not to respond to the search task, but to attend and memorize these orientations. Then a horizontal line and a response wheel were presented at the center of the to-be-reported item, remained on until the response. Participants had to rotate the line and clicked the left key on the mouse when they felt the orientation was the same as the one presented in the probe display. Responses were unspeeded and only accuracy was emphasized. The inter-trial interval (ITI) was between 500 and 750 ms at random.

The search target was presented on each trial, and it was equally likely to be a circle or a diamond. A uniquely colored distractor singleton was randomly presented in 66% of the trials in each block, with the same shape as other distractors but a different color (red or green balanced between subjects). One of these distractor locations had a high proportion of 62.5% (high-probability location), and other locations shared a low proportion of 37.5% with each had a low probability of 7.5% (low-probability location). The high-probability location remained the same for each participant and was counterbalanced across participants. In the condition with a distractor the target never appeared at the high-probability location, but appeared equally often at all other locations. This design adopted was the same as in Wang and Theeuwes [1]. One might question whether the effect reported in the present study was due to the fact that the target was never presented at the high-probability distractor location or whether it has nothing to do with target probability but instead is completely due to the fact that the distractor was presented at that location much more often. A recent study by Failing et al., answered exactly this question and showed that the suppression is solely due to the probability of that the distractor is presented at that location [9]. Participants were first trained for 360 trials to understand the search task before the testing. Then, they completed 40 practice trials and 7 blocks with each containing 360 trials in two successive days (a total of 2520 trials), in which search-only and search-probe trials were mixed within blocks.

## Additional analysis

For search-probe trials, a standard mixture model was fitted to characterize the distribution of response errors in terms of response precision and guess rate [42]. The response error was calculated by subtracting the correct value of probed orientation from the response value. The distribution was assumed to consist of a uniform distribution of response errors for guessing trials and a von Mises (circular normal) distribution of response errors for non-guessing trials. By using maximum likelihood estimation, the distribution of the response error data from each condition was entered into the model,

$$P(e) = (1 - g)\Phi_\sigma(e) + g/2\pi,$$

Where one input parameter $e$ (response errors) is required, and two output parameters $g$ (guess rate, the proportion of the guess trials) and $\sigma$ (standard deviation [SD], the width of the von Mises distribution, reflecting the precision of the internal representation) will be given. The MemToolbox was used to fit the current dataset [47].

## Results

### Search-only condition

Trials (2.1%) on which the response times (RTs) were slower than 1500 ms or faster than 200 ms were removed from analysis.

**Attentional capture effect.** Mean RTs and mean error rates are presented in Fig 2A. With *distractor condition* (high-probability location, low-probability location, and no-distractor) as a factor, a repeated measures ANOVA on mean RTs showed a main effect, $F(2, 30) = 159.22$, $p < .001$, $\eta_p^2 = .91$. Subsequent planned comparisons showed that, against the no-distractor condition, there were significant attentional capture effects when the distractor singleton was presented at the high-probability location, $t(15) = 11.25$, $p < .001$, *cohen's d* = 0.43, and when it was presented at the low-probability location, $t(15) = 14.01$, $p < .001$, *cohen's d* = 1.14.

## A) Different distractor conditions

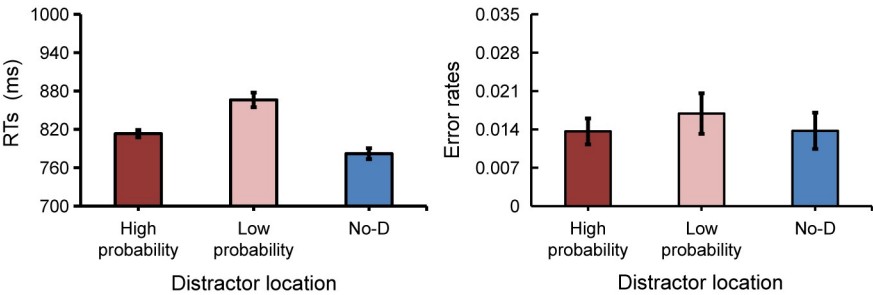

## B) Distractor singleton absent

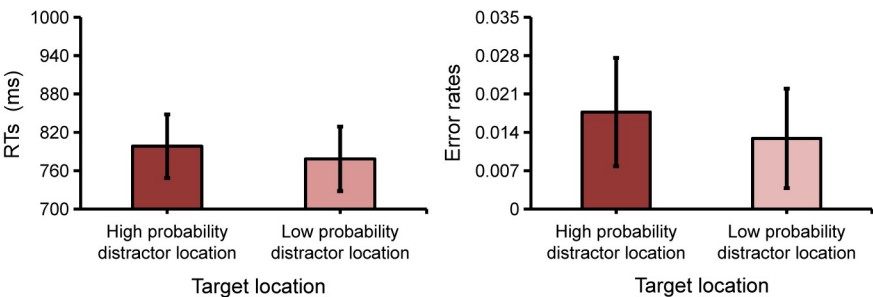

## C) Spatial distribution

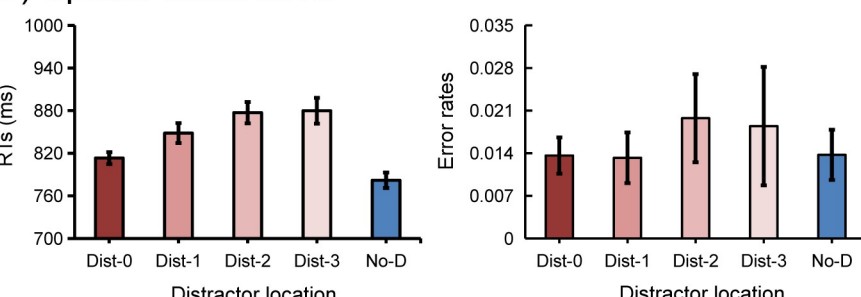

**Fig 2. Results of *search-only* trials.** The mean response times (RTs) and mean error rates in different distractor conditions (A) and in the distractor singleton absent condition (B). The spatial distribution of attentional capture effect by the means of response times and error rates in the distractor singleton present condition (C). Here, Dist-0 represents the high-probability distractor location, Dist-1 represents the low-probability distractor location with 60˚ polar angle away from the high-probability distractor location (physical distance), and so on. Error bars denote 95% confidence intervals (CIs).

Consistent with Wang and Theeuwes [1, 14–15], the difference between high- and low-probability locations was also reliable, $t(15) = 10.71$, $p < .001$, *cohen's d* = 0.72, suggesting that the attentional capture effect was attenuated for trials in which the distractor singleton appeared at the high-probability location (see S1 Appendix for training results). Importantly, when analyzing different blocks separately, we found that the suppression effect already occurred in the first block, $t = 8.49$, $p < .001$, and remained present in the following blocks, all $ps < .003$, suggesting that learning to suppress the high-probability location is very efficient. No such effect was observed on error rates, $F(2, 30) = 2.32$, $p = .116$, $\eta_p^2 = .13$, BF01 = 1.32.

**Target selection.** Mean RTs and mean error rates are presented in Fig 2B. To further examine the efficiency of target selection we calculated the mean RTs in the distractor absent condition. Pairwise t-test showed that the selection was less efficient when the target was presented at the high-probability location compared to when it was presented at the low-probability location, $t(15) = 2.54$, $p = .023$, *cohen's d* = 0.26. There was no effect on error rates, $t(15) = 1.18$, $p = .258$, *cohen's d* = 0.3, BF01 = 2.17.

**The spatial distribution of the suppression effect.** To explore the spatial gradient of the suppression effect, we divided the distractor locations into four distances (dist-0, dist-1, dist-2, and dist-3). The mean RTs and mean error rates for these conditions are presented in Fig 2C. A one-way repeated measures ANOVA with distance as a factor showed a significant main effect for RTs, $F(4, 60) = 54.62$, $p < .001$, $\eta_p^2 = .79$, but not for error rates, $F(4, 60) = 1.42$, $p = .237$, $\eta_p^2 = .09$. We fitted the RT data with a linear function (as one of the options to capture the gradient of the suppression effect [1]) and used its slope to describe the decrease of the suppression effect with the increase of the distance relative to the high-probability location. The slope (23.01 ms per display element) for mean RTs was significantly larger than zero, $t(15) = 6.7$, $p < .001$, *cohen's d* = 2.37, suggesting that the suppression effect was not limited to one location, but had an extended spatial gradient.

## Search-probe condition

**Response error.** The mean response errors are presented in Fig 3A. To examine the impact of statistical learning on the spatial distribution of attention, we sorted the distractor singleton present condition into two conditions: distractor present at the high-probability location and at the low-probability location. A new condition, named *probe type* (target, neutral-element, distractor-singleton), was defined as well. It denotes that the probe could be presented at target location, neutral-element location, or distractor-singleton location. A repeated-measures ANOVA on mean response errors with factors of *distractor location* (high-probability location vs. low-probability location) and *probe type* (target, neutral-element, distractor-singleton) showed a significant main effect for probe type, $F(2, 30) = 26.92$, $p < .001$, $\eta_p^2 = .64$, but not for distractor location, $F(1, 15) = 0.18$, $p = .674$, $\eta_p^2 = .01$, BF01 = 2.05×10^10. Importantly, we observed a significant interaction between distractor location and probe type, $F(2, 30) = 5.42$, $p = .01$, $\eta_p^2 = .27$. To unpack the main effect of probe type, we performed subsequent t-tests. When the probed item was presented at the target location, the performance was superior compared to that when the probed item was presented at the neutral-element location, $t(15) = 6.94$, $p < .001$, *cohen's d* = 1.31, and at the distractor-singleton location, $t(15) = 5.12$, $p < .001$, *cohen's d* = 1.1. When the probe was presented at the distractor-singleton location, performance was better compared to that when the probe was presented at the neutral-element location, $t(15) = 2.36$, $p = .032$, *cohen's d* = 0.12.

Subsequent comparisons showed that when the probed item was presented at the distractor-singleton location, the performance was worse for distractor singletons that appeared at the high-probability location than at the low-probability location, $t(15) = 2.19$, $p = .045$,

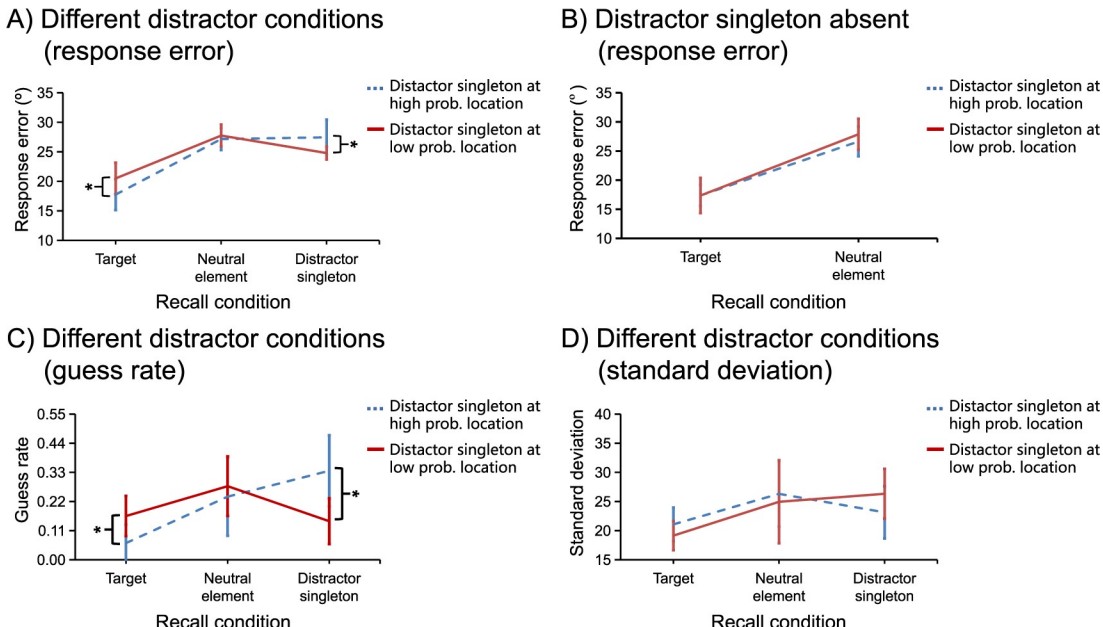

**Fig 3. Results of *search-probe* trials.** The mean response errors in different distractor conditions (A) and in the distractor singleton absent condition (B). The mean guess rates (C) and mean standard deviation (D) in different distractor conditions. Error bars denote 95% CIs.

*cohen's d* = 0.29, suggesting that the high-probability location was suppressed relative to the low-probability location. Consistently, we also found that when the probed item was presented at the target location, the suppression pattern was reversed; i.e., the performance was now better for distractors presented at the high-probability location than at the low-probability location, $t(15) = 3.42$, $p = .004$, *cohen's d* = 1.79, suggesting that the processing of the target was facilitated due to more suppression when the distractor singleton was presented at the high-probability location relative to the low-probability location. When the probed item was presented at any of the neutral-element locations (not at the target nor at the distractor singleton location), there was no difference in performance between the distractor presented at the high- vs. low-probability location, $t(15) = 0.5$, $p = .626$, *cohen's d* = 0.07, BF01 = 3.51. This suggests that the distractor suppression does not affect the processing of neutral elements.

In the no-distractor condition, a two-way ANOVA was conducted on mean response errors as well, with factors of *recall location* (high-probability location vs. low-probability location) and *probe type* (target vs. neutral-element). The results showed a significant main effect for probe type, $F(1, 15) = 46.0$, $p < .001$, $\eta_p^2 = .75$, but not for recall location, $F(1, 15) = 0.68$, $p = .422$, $\eta_p^2 = .04$, BF01 = $2.06 \times 10^{10}$; and there was no interaction, $F(1,15) = 0.75$, $p = .401$, $\eta_p^2 = .05$, BF01 = 2.66. The performance was better when the probe was presented at the target location than at the neutral-element location (see Fig 3B).

We also examined the difference in mean response errors for probing the target location between the no-distractor condition and the condition when the distractor was presented at the high- and low-probability location. The results showed that, when the probe was presented at the target location, there was no difference between the no-distractor condition and the condition that the distractor was presented at the high-probability location, $t(15) = 0.51$, $p = .618$, *cohen's d* = 0.07, BF01 = 3.5. However, when the distractor was presented at the low-probability location, the mean response errors for probing the target location was significantly larger than in the no-distractor condition, $t(15) = 3.57$, $p = .003$, *cohen's d* = 0.47.

**Guess rate.** The mean guess rates are presented in Fig 3C. A repeated-measures ANOVA on mean guess rates with factors of *distractor location* (high-probability location vs. low-probability location) and *probe type* (target, neutral-element, distractor-singleton) showed a significant main effect for probe type, $F(2, 30) = 6.66$, $p = .004$, $\eta_p^2 = .31$, but not for distractor location, $F(1, 15) = 0.17$, $p = .69$, $\eta_p^2 = .01$, BF01 = 35.8. When the probed item was presented at the target location, the guess rate was lower compared to that when the probed item was presented at the neutral-element location, $t(15) = 3.01$, $p = .009$, *cohen's d* = 0.78, and at the distractor-singleton location, $t(15) = 2.63$, $p = .019$, *cohen's d* = 0.69. Importantly, we observed a significant interaction between distractor location and probe type, $F(2, 30) = 4.6$, $p = .018$, $\eta_p^2 = .24$.

Subsequent comparisons showed that when the probed item was presented at the distractor-singleton location, the guess rate was higher for distractor singletons that appeared at the high-probability location than at the low-probability location, $t(15) = 2.54$, $p = .023$, *cohen's d* = 0.7. We also found that when the probed item was presented at the target location, the pattern was reversed; i.e., the guess rate for the target was lower when the distractor singleton was presented at the high- relative to the low-probability location, $t(15) = 2.77$, $p = .014$, *cohen's d* = 0.68. When the probed item was presented at any of the neutral element locations, there was no difference between distractor singletons present at the high-probability location and at the low-probability location, $t(15) = 0.44$, $p = .669$, *cohen's d* = 0.14, BF01 = 3.6. Clearly, the results of guess rate mimic what we have found for response errors.

**Standard deviation.** The mean SDs are presented in Fig 3D. A repeated-measures ANOVA on mean SDs with factors of *distractor location* (high-probability location vs. low-probability location) and *probe type* (target, neutral-element, distractor-singleton) showed a significant main effect for probe type, $F(2, 30) = 5.41$, $p = .01$, $\eta_p^2 = .27$, but not for distractor location, $F(1, 15) < 0.01$, $p = .982$, $\eta_p^2 < .01$, BF01 = 12.15. When the probed item was presented at the target location, the SD was lower compared to that when the probed item was presented at the neutral-element location, $t(15) = 2.73$, $p = .015$, *cohen's d* = 0.81, and at the distractor-singleton location, $t(15) = 3.38$, $p = .004$, *cohen's d* = 0.69. However, there was no interaction, $F(2, 30) = 0.7$, $p = .507$, $\eta_p^2 = .04$, BF01 = 3.22.

## Discussion

The current study replicated precisely all previous findings of Wang and Theeuwes [1]. We showed that for the high-probability location there were (1) less capture by the salient distractor and (2) less efficient selection of the target. There was also a spatial gradient from the high-probability location as the attentional capture effect scaled with the distance from this location.

In addition to this replication, the search-probe condition elegantly demonstrates the initial distribution of attentional resources across the visual field. Consistent with a space-based resource allocation ("biased competition") model we showed that the high-probability location was suppressed relative to the low-probability location as there were more response errors and higher guess rate in the high- relatively to the low-probability location. At the same time, this suppression of the high-probability location resulted in more attention being allocated to the target location relative to a condition in which the distractor was not proactively suppressed, i.e., when presented at a low-probability location.

The current findings are consistent with notion that this type of learning results in *proactive* suppression. Indeed, because the high-probability location is proactively suppressed (i.e., before display onset) this location competes less for attention than the other locations, giving rise to more response errors for probes presented at the high-probability location than at the low-probability location. If suppression would have been applied after attention has initially

shifted there (so called retro-active suppression), then we would have expected that probes would have been picked up at the high-probability location just as well as any other location within short time window. Several speculations might be derived from the results in the search-probe task. It seems to suggest that the reduction in interference of a salient distractor when presented at a high-probability location is the result of a combination of less capture by the distractor and *at the same time* more attentional allocation to the target. Similarly, when a distractor is presented at the low-probability location, the strong attention capture observed summons so much attention that less attention is available for target processing.

It should be noted that our analysis can say little, if anything, about whether attention is allocated in parallel or serially. The search display was presented for 200 ms and previous research has shown that within this time window, attention is first summoned by the salient distractor before it is allocated to the target. For example, Kim and Cave [48] used Theeuwes' additional singleton paradigm with only 4 items in the display and combined it with a probe detection task. When the probe was shown 60 ms after the display onset, observers responded 20 ms faster when a probe was presented at the distractor location relative to the target location. At 150 ms interval, this pattern was reversed: the mean RTs at the target location was about 15 ms faster than at the distractor location. Kim and Cave [48] argued that, at 60 ms after display onset, more attention was allocated at the distractor location than at the target location signifying attentional capture. Soon thereafter (at 150 ms condition) attention was disengaged from the distractor location and directed at the target location. Even though we cannot take the exact timings reported by Kim and Cave as absolute (there were many differences between our singleton task and theirs, e.g., the number of items in the display), it is likely that within a short time period of 200 ms, attention may have shifted between distractor and target locations.

Our finding that the mean response errors for probing the target location in the no-distractor condition did not differ from that when the distractor was presented at the high-probability location, suggests that the distractor presented at the high-probability location hardly competes for attention. This analysis suggests that due to proactive suppression, there are equivalent processing resources available for target processing as there are in a condition in which the distractor is not present (i.e., the no-distractor condition), a finding consistent with the notion that this type of suppression is *proactive in nature* [37].

Note that we employed here a version of the additional singleton task [49] in which the target and distractors switched roles across trials. When using this version, participants likely employ the so-called singleton detection mode which may in turn result in stronger capture effects than when having observers search consistently for one specific feature [so called "feature search mode", see 50; but see 51]. We took the former approach to examine the interplay between bottom-up capture and statistical learning minimizing top-down effects on search. Note however that even when one uses displays that induce feature search, the same suppression effect is observed indicating that this type of suppression does not depend on the search mode employed [15].

## Supporting information

**S1 Appendix.**
(DOCX)

## Author Contributions

**Data curation:** Siyang Kong.

**Methodology:** Siyang Kong, Benchi Wang.

**Supervision:** Xinyu Li, Benchi Wang.

**Writing – original draft:** Siyang Kong, Xinyu Li, Benchi Wang, Jan Theeuwes.

**Writing – review & editing:** Benchi Wang, Jan Theeuwes.

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
