## [Decision Letter · Decision Letter 0]

18 Feb 2020

PONE-D-19-35559

Proactively location-based suppression elicited by statistical learning

PLOS ONE

Dear Mr. Wang,

Thank you for submitting your manuscript to PLOS ONE. After careful consideration, we feel that it has merit but does not fully meet PLOS ONE’s publication criteria as it currently stands. Therefore, we invite you to submit a revised version of the manuscript that addresses the points raised during the review process.

The reviewers all indicated that your work was technically sound and analyzed appropriately and, on the whole, their assessments are quite positive. However, all three raised important points which require further clarification in the manuscript, some of which may require changes to the data analyses (or at least a stronger defense of the analysis decisions that you have made). I note that some of the same concerns were raised by multiple reviewers, for instance the relevance of probability learning about the target location as a relevant factor that ought to be discussed, and the possibility of mapping out an acquisition function across the multiple blocks of testing. As the reviewers’ points are clear, I won’t reiterate all of them here, but I would expect to see each concern addressed either in revisions to the manuscript or in rebuttal before this work can be deemed acceptable for publication. 

I will note that, consistent with several of the reviewers’ concerns, I too struggled to find some of the relevant methodological details. For instance, it would be desirable to know how the distractor and no-distractor conditions were intermixed within blocks and what proportion of trials were no-distractor trials (perhaps this information is buried in there but it was not obvious to me). Perhaps Figure 1 could also be referred to earlier, for example page 7 when you introduce the present study. In addition, the Abstract currently provides very little context for readers who are not familiar with the additional singleton task. Given the wide readership of the journal, you might consider softening the blow, so to speak.

We would appreciate receiving your revised manuscript by Apr 03 2020 11:59PM. To enhance the reproducibility of your results, we recommend that if applicable you deposit your laboratory protocols in protocols.io, where a protocol can be assigned its own identifier (DOI) such that it can be cited independently in the future. For instructions see: http://journals.plos.org/plosone/s/submission-guidelines#loc-laboratory-protocols

We look forward to receiving your revised manuscript.

Kind regards,

Evan James Livesey, Ph.D

Academic Editor

PLOS ONE

Journal Requirements:

1. We noticed you have some minor occurrence of overlapping text with the following previous publication(s), which needs to be addressed:

https://link.springer.com/article/10.3758%2Fs13414-018-1562-3

https://www.mitpressjournals.org/doi/full/10.1162/jocn_a_01433

https://link.springer.com/article/10.3758%2Fs13423-019-01679-6

https://www.sciencedirect.com/science/article/pii/S2352250X18301970?via%3Dihub

In your revision ensure you cite all your sources (including your own works), and quote or rephrase any duplicated text outside the methods section. Further consideration is dependent on these concerns being addressed.

Reviewers' comments:

Reviewer's Responses to Questions

**Comments to the Author**

1. Is the manuscript technically sound, and do the data support the conclusions?

Reviewer #1: Yes

Reviewer #2: Yes

Reviewer #3: Yes

2. Has the statistical analysis been performed appropriately and rigorously? 

Reviewer #1: Yes

Reviewer #2: Yes

Reviewer #3: Yes

3. Have the authors made all data underlying the findings in their manuscript fully available?

Reviewer #1: Yes

Reviewer #2: Yes

Reviewer #3: Yes

4. Is the manuscript presented in an intelligible fashion and written in standard English?

Reviewer #1: Yes

Reviewer #2: Yes

Reviewer #3: Yes

5. Review Comments to the Author

Reviewer #1: PONE-D-19-35559: Proactively location-based suppression elicited by statistical learning

This study replicated the previous findings that attentional capture is reduced by salient distractor when it appears in high probability locations of distractor compared to when it appears in low probability locations of distractor. Also, this study tested how attentional resources are distributed across the visual field by adding a search-probe condition. The authors showed that high probability location is suppressed before display onset and suggested that the suppression of high probability location resulted in more attention being allocated to the target location.

Review: This manuscript is very clearly written and had an excellent summary of statistical learning literature and the authors' previous findings in the introduction. Although I have some comments that might require additional data analysis and questions that need clarification, I believe this manuscript will be an excellent addition to the journal, PLOS ONE.

Comment #1. The authors informed that the target appeared equally often at each location in the no-distractor condition but did not report how the target was distributed in the distractor condition. It has been well known that target probability guides spatial attentional - high probable location of target quickly and implicitly attracts spatial attention (Geng & Bermann 2002; 2005; Jiang et al., 2013). I wonder, in this study, if the high probability location of color singleton contained the target less often than other locations, which might result in reducing attentional capture to that location. I hope the authors clarify this point in the revision.

Comment #2: I wonder if the search performance in the high probability location gradually increased across seven testing blocks or already reached to the ceiling when testing began. Considering that the effect came from statistical "learning," many readers will wonder about the learning curve.

Comment #3: In search-probe results, it is interesting that the response error of target location in high probability condition did not differ from that in the no-distractor condition because the search performance was much worse in the former condition than in the latter. I hope the authors provide some explanations about this discrepancy.

1. Typos: page 3, line 9: "categories" to "category"; page 5, line 15: "RT an accuracy" to "RT and accuracy"

2. Redundant sentence: page 15, lines 6-7: "Importantly, there was also a ..." is redundant (it has been very similarly mentioned on page 14, line 19)

3. Inconsistent BF report: It is a little odd that the authors provided only one BF value (page 16, the very bottom line). Please add all BFs if possible and interpret the meaning.

4. I believe the authors meant "former (singleton detection mode)" not "latter (feature search mode)" on page 19, line 14.

Reviewer #2: SUMMARY

The authors examined the influence of statistical learning on the capture of spatial attention by combining a search-probe task with a visual-search paradigm (Wang and Theeuwes (2018a)). For the majority of trials, participants only completed the visual-search: participants searched for a shape singleton, sometimes in the presence of a color singleton distractor. Critically, one of the locations had a higher probability of containing the distractor than the other locations. To engage implicit statistical learning processes, participants were not informed of this relation. On the rest of the trials, the search display was quickly interrupted by a memory probe display where the items were replaced by bars of different orientations. Participants were required to memorize the orientations of the bars before being probed to recall the orientation at one location.

The authors report significantly slower reaction times when the distractor color singleton appeared, with more capture occurring when the distractor was in the low-probability position than in the high-probability position. Error rates were also significantly higher when the distractor singleton was presented at the low-probability location but not when the distractor singleton was presented at the high-probability location. Reaction times were also slower when the target was in the high-probability location but not the low-probability location.

In the search-probe task, memory performance, measured by mean response error, was best when the probe was at the target location of the search display compared to the neutral or distractor locations. Recall when the distractor location was probed was worse at the high-probability location compared to the low-probability location, but when the probed location was the search target location, recall was more accurate when the distractor was at the high-probability location compared to the low-probability location. The authors interpret these results as suppression of the high-probability distractor location and also more attentional allocation to the target.

MAIN ISSUES

1. In the analysis of the search-probe condition, an improvement to the mean response error may be to use mixture modelling like those in visual working memory recall tasks. Fitting a combination of a Von Mises distribution and uniform distribution to the response errors will give two parameters: the precision of the recall and the amount of guessing in the recall. These parameters might be more sensitive than mean response error because it is possible that while more guessing is occurring when the distractor is at the low-probability location the mean response error remains centered at the same value. Given the brief display time of the probe arrays, it is likely that the proportion of memory responses (those in the Von Mises distribution) is larger when the distractor is in the high-probability location compared to the low-probability location.

2. There are two aspects of statistical learning that appear to be relevant; it is automatic and implicit, such that it lends itself to bottom-up rather than top-down effects. Was awareness of the statistical regularities tested in this experiment? While the procedure may previously have been shown to exist without awareness of the regularities, it might be necessary to examine whether the effects of attentional capture vary between people who are explicitly aware of the regularities or not. Given the size of some of the interaction effects, it might also be necessary to check whether these effects stay the same excluding those with explicit awareness of the statistical regularities.

3. There are some instances where it isn’t clear in the procedure whether only one other location was a low-probability location, or all other locations were low-probability. Below is an example of where it is confusing whether one low-probability location or multiple low-probability locations were used:

P11. “One of these distractor locations had a high proportion of 62.5% (high-probability location), and other locations had a low proportion of 37.5% (low-probability location).”

P12. “When the distractor singleton was presented at the low-probability location, but not when it was presented at the high-probability location.”

4. In examining the spatial distribution of the suppression effect (pp. 13-14), it looks like a linear function doesn’t best describe what is happening in Figure 2C. Perhaps a polynomial regression with linear and quadratic components might explains the trend better, suggesting moreso that the suppression is at the high-probability location and the two adjacent locations, rather than spreading linearly along the gradient.

5. For the contrast and post-hoc comparisons throughout the results, was any error rate correction applied? If so, that should be clarified in the description of the results. For example, the follow-up t-tests (pp.14-15) may no longer be significant if a correction needs to be applied. If these contrasts were planned, it would need to be mentioned too.

6. Exploring the time-course of attentional suppression could be fruitful, achieved by delaying the onset of the probe array. You might expect that if attention is captured by the salient distractor, you could observe recovery from the attentional capture with longer delays before presenting the memory array. However, a sustained suppression account might suggest that the effect does not change with a delayed onset of the memory array.

MINOR ISSUES

1. It might be helpful to refer to the probe array as a memory array to differentiate between the visual search task.

2. I think the discussion of the proactive suppression in the discussion might benefit with contrast to what is expected with retroactive suppression.

3. It might be helpful to add to the Procedure where the experimental code, data and analysis code may be accessed to the manuscript.

TYPOGRAPHICAL ERRORS

pp. 3. “Recently, it was pointed out that a third categories”, should be “a third category”.

pp. 7. The reference is missing a comma, “Wang, van Driel, et al. 2019).

pp. 10. “did not respond or respond incorrectly” should be “or responded incorrectly”.

pp. 12. “The F-value of the second paragraph has an extra period.

pp 12. “Again, there was significantly different” should be “there was a significant difference between…”

pp. 12. “Paired-wise t-test” should be “Pairwise t-tests”

William Xiang Quan Ngiam

I sign all my reviews, regardless of the recommendation to the editor. By signing this review, I affirm that I have made my best attempt to be polite and respectful while providing criticism and feedback that is hopefully helpful and reasonable.

Reviewer #3: The manuscript details a single experiment examining attentional capture and suppression of locations based on the statistical properties of the task-environment. They use a classic additional singleton paradigm, in which search for a uniquely shaped object is impaired in the presence of a unique colour singleton distractor, showing the classic effect of slower RTs to targets on these trials compared to trials in which that distractor is absent. Here though, the effect of the distractor is modulated by the probability with which it appears in certain locations, with the distractor being less effective when it appears in a high- compared to a low-probability location. This suggests that participants are learning to suppress these high-probability locations, and provides a replication of effects previously reported by Wang and Theeuwes (2018). The novel contribution here is the introduction of a new procedural element: on a third of trials, a probe test is given in which each location in the search array is quickly masked by slanted lines, and then participants are tasked with reporting on one of these. The accuracy of this report reveals interesting things about the momentary suppression of the spatial location: when a singleton distractor appears at a high probability location, participants are worse at reporting the probe in that location, and better at reporting the probe at the target location, relative to the when the singleton distractor appears at a low-probability location.

In general I thought this was an interesting set of data and should make a nice contribution to the literature. I had a number of suggestions as to how I thought the authors could improve the manuscript. Points 1-3 I see as fairly critical, but I had a number of more minor points.

1. The presentation of the rationale for the paper currently makes it seem like the contribution here is fairly modest. That is, similar effects have already been reported in the Wang and Theeuwes (2018) papers, and it wasn’t immediately clear what is left unanswered by those papers, and what the current paper addresses. I appreciate that the current paper provides a more direct test of suppression at various positions with the probe task, but I think that novel contribution, and its importance, can be made more clear in the introduction for those less familiar with the methods and results of previous papers.

2. It isn’t clear exactly what the competing hypothesis is for this experiment. On page 6 the authors state that “Proactive suppression can be contrasted with retroactive suppression, which is the type of suppression that occurs only after attention has been directed to a location, disengaged and subsequently suppressed.” Could the authors expand upon this sentence and flesh out the range of possible results they considered and the theories they would support? I feel that this very brief treatment of the theoretical possibilities weakened the manuscript and it wasn’t clear how the current data sit with such alternative accounts. For example, consider the data shown in Figure 3B. If I’ve interpreted this correctly, these data suggest that in the absence of a distractor singleton, there is no impairment in probe responding at the high probability location relative to the low probability location. Doesn’t that suggest that it is indeed the presence of the highly salient singleton that attracts attention, and it’s only when that occurs, that active suppression comes into play? That sounds similar to what you describe above for “retroactive suppression”, but I didn’t see that mentioned in the discussion.

3. Given that this is an effect of learning the statistical properties of the spatial locations, and given that each participants contributed 2520 trials, it seemed to me a missed opportunity not to analyse how these patterns develop over time. Could the authors break the data down into blocks of trials so we can see the development of the effect?

4. The description of the contextual cuing literature on page 4 doesn’t seem very accurate. Firstly, the statement “…participants were faster in finding targets when they appeared in repeated configurations than in novel locations” is ambiguous, in that “novel locations” could refer to the target location. Better would be “…participants were faster to find targets when they appeared in repeated configurations than when they appeared in novel configurations”. Secondly, I don’t agree that this is related to Posner cueing tasks, at least not in the way that is stated in the next sentence. Contextual cuing is generally not about learning the probability of where targets appear per se. Rather, it is about learning the association between configuration and target. Typically target location probability is controlled across repeated and random configurations.

5. Having said that, there is a small literature on probability cuing in contextual cuing of visual search, which starts with the paper by Jiang et al. (2013). This will be of interest to the authors and may be relevant for the discussion here.

6. pg. 9 – it wasn’t initially clear to me from Figure 1 how the participants knew which probe was being queried in the task. It is mentioned in the text briefly, but I found the figure a bit confusing, given the final panel presumably shows a magnified/schematic version of the screen. I think this Figure needs some work to clarify how the task operated, perhaps showing both the probe recall prompt in one of the actual positions, and then also magnified to show the detail.

7. I appreciate that only a small percentage of trials were removed as outliers, but I would prefer to see the upper limit on RTs be set on a participant basis and related to the distribution of RTs for that participant (e.g., 2.5/3 SDs above the mean RT). Otherwise, this is set fairly arbitrarily and it begs the question of why this value was selected. Were the results in any way contingent upon this upper threshold value?

8. I applaud the authors for making their materials and data available. I downloaded these and I was able to read in the data files, but I couldn’t make much sense of many of the variables in the data files. I would suggest including a document that lays out clearly what the variables are, what the levels of those variables are coded as, etc. It would be great if the authors also included their analysis scripts.

Jiang, Y. V., Swallow, K. M., & Rosenbaum, G. M. (2013). Guidance of spatial attention by incidental learning and endogenous cuing. Journal of Experimental Psychology: Human Perception and Performance, 39, 285–297.

6. PLOS authors have the option to publish the peer review history of their article (what does this mean?). If published, this will include your full peer review and any attached files.

Reviewer #1: No

Reviewer #2: Yes: William Xiang Quan Ngiam

Reviewer #3: No

---

## [Author Response · Author response to Decision Letter 0]

22 Mar 2020

Dear Dr. Evan James Livesey,

We would like to thank you and the three reviewers for their excellent comments and suggestions. We took their comments to heart and changed the manuscript accordingly. All the changes were marked in red in Ms.

Hope that the current version is now acceptable. Below we outline how we addressed each concern of you and the reviewers.

Thank you very much!

Sincerely

Benchi Wang & Jan Theeuwes

Editor

I will note that, consistent with several of the reviewers’ concerns, I too struggled to find some of the relevant methodological details. For instance, it would be desirable to know how the distractor and no-distractor conditions were intermixed within blocks and what proportion of trials were no-distractor trials (perhaps this information is buried in there but it was not obvious to me). 

Fixed. We have made it clear, see p. 11.

Perhaps Figure 1 could also be referred to earlier, for example page 7 when you introduce the present study. In addition, the Abstract currently provides very little context for readers who are not familiar with the additional singleton task. Given the wide readership of the journal, you might consider softening the blow, so to speak.

Fixed.

Reviewer #1 

This manuscript is very clearly written and had an excellent summary of statistical learning literature and the authors' previous findings in the introduction. Although I have some comments that might require additional data analysis and questions that need clarification, I believe this manuscript will be an excellent addition to the journal, PLOS ONE.

Thanks for your comments.

Point 1. The authors informed that the target appeared equally often at each location in the no-distractor condition but did not report how the target was distributed in the distractor condition. It has been well known that target probability guides spatial attentional - high probable location of target quickly and implicitly attracts spatial attention (Geng & Bermann 2002; 2005; Jiang et al., 2013). I wonder, in this study, if the high probability location of color singleton contained the target less often than other locations, which might result in reducing attentional capture to that location. I hope the authors clarify this point in the revision.

Response: Thanks for pointing this out. Yes, the target never appeared in the high-probability location in the with-distractor condition. However, in a recent study, we have shown that the target probability is not responsible for the effect (Failing, Wang, et al., 2019). That is, even when the target is equally likely presented at all locations, the suppression is seen for the high-probability distractor location. Thus, we kept adopting the same design as in Wang & Theeuwes (2018a). We have made it clear in the Ms, see p. 11.

Point 2: I wonder if the search performance in the high probability location gradually increased across seven testing blocks or already reached to the ceiling when testing began. Considering that the effect came from statistical "learning," many readers will wonder about the learning curve.

Response: Thanks. We now report it in the Ms, see p. 13. Similar to what we have observed in previous studies, people learned to suppress the high-probability location very quickly. The results showed that the mean RTs for the high-probability location was smaller than that for low-probability location in the first block, p < .001, and in the following blocks, all ps < .003.

Point 3: In search-probe results, it is interesting that the response error of target location in high probability condition did not differ from that in the no-distractor condition because the search performance was much worse in the former condition than in the latter. I hope the authors provide some explanations about this discrepancy.

Response: Thanks. We thought we explained it in the general discussion in p. 22, see below:

“Our finding that the mean response errors for probing the target location in the no-distractor condition did not differ from that when the distractor was presented at the high-probability location, suggests that the distractor presented at the high-probability location hardly competes for attention. This analysis suggests that due to proactive suppression, there are equivalent processing resources available for target processing as there are in a condition in which the distractor is not present (i.e., the no-distractor condition), a finding consistent with the notion that this type of suppression is proactive in nature (see also Wang, van Driel, et al., 2019).” 

Minor

 1. Typos: page 3, line 9: "categories" to "category"; page 5, line 15: "RT an accuracy" to "RT and accuracy"

2. Redundant sentence: page 15, lines 6-7: "Importantly, there was also a ..." is redundant (it has been very similarly mentioned on page 14, line 19)

3. Inconsistent BF report: It is a little odd that the authors provided only one BF value (page 16, the very bottom line). Please add all BFs if possible and interpret the meaning.

4. I believe the authors meant "former (singleton detection mode)" not "latter (feature search mode)" on page 19, line 14.

Response: All fixed. We have reported BF values for all null results.

Reviewer #2

Thanks for your comments.

Point 1. In the analysis of the search-probe condition, an improvement to the mean response error may be to use mixture modelling like those in visual working memory recall tasks. Fitting a combination of a Von Mises distribution and uniform distribution to the response errors will give two parameters: the precision of the recall and the amount of guessing in the recall. These parameters might be more sensitive than mean response error because it is possible that while more guessing is occurring when the distractor is at the low-probability location the mean response error remains centered at the same value. Given the brief display time of the probe arrays, it is likely that the proportion of memory responses (those in the Von Mises distribution) is larger when the distractor is in the high-probability location compared to the low-probability location.

Response: Excellent suggestion. We have put the model analysis in.

Point 2. There are two aspects of statistical learning that appear to be relevant; it is automatic and implicit, such that it lends itself to bottom-up rather than top-down effects. Was awareness of the statistical regularities tested in this experiment? While the procedure may previously have been shown to exist without awareness of the regularities, it might be necessary to examine whether the effects of attentional capture vary between people who are explicitly aware of the regularities or not. Given the size of some of the interaction effects, it might also be necessary to check whether these effects stay the same excluding those with explicit awareness of the statistical regularities.

Response: We did not test awareness in the current study. However, we did it in several previous studies (e.g., Wang & Theeuwes, 2018a 2018b), and found that most people were basically unware of the regularities, even though they learned to suppress the distractor. This suggests that learning is basically implicit. We mentioned the previous results in Ms, see p. 5.

We do agree with that this is an interesting question, yet given by that we did not test it so that we keep not mention it in the current Ms. 

Point 3. There are some instances where it isn’t clear in the procedure whether only one other location was a low-probability location, or all other locations were low-probability. Below is an example of where it is confusing whether one low-probability location or multiple low-probability locations were used:

P11. “One of these distractor locations had a high proportion of 62.5% (high-probability location), and other locations had a low proportion of 37.5% (low-probability location).”

P12. “When the distractor singleton was presented at the low-probability location, but not when it was presented at the high-probability location.”

Response: We have explained it, see p. 11. 

Point 4. In examining the spatial distribution of the suppression effect (pp. 13-14), it looks like a linear function doesn’t best describe what is happening in Figure 2C. Perhaps a polynomial regression with linear and quadratic components might explains the trend better, suggesting moreso that the suppression is at the high-probability location and the two adjacent locations, rather than spreading linearly along the gradient.

Response: We agree with that, as shown in Figure 2, it seems that only the high-probability location and the two adjacent locations were suppressed. However, even though we used other fitting functions, it still suggests that the suppression effect was not limited to one location, but had an extended spatial gradient, in line with our claim. To be consistent with previous studies, we kept using liner function to describe the trend of the spatial gradient. We do not make strong claims about whether it is truly a linear function. see p. 15.

Point 5. For the contrast and post-hoc comparisons throughout the results, was any error rate correction applied? If so, that should be clarified in the description of the results. For example, the follow-up t-tests (pp.14-15) may no longer be significant if a correction needs to be applied. If these contrasts were planned, it would need to be mentioned too.

Response: Thanks. These comparisons were all planned comparison (based on the results of our previous studies). 

Point 6. Exploring the time-course of attentional suppression could be fruitful, achieved by delaying the onset of the probe array. You might expect that if attention is captured by the salient distractor, you could observe recovery from the attentional capture with longer delays before presenting the memory array. However, a sustained suppression account might suggest that the effect does not change with a delayed onset of the memory array.

Response: We agree with that a more fined-grained exploration of the time course would be truly interesting. This would require another study. Regarding this study we argue that within a short time period, attention shifts between distractor and target locations.

Minor 

1. It might be helpful to refer to the probe array as a memory array to differentiate between the visual search task.

We kept the term probe array even though we realize that it is also a memory array.

2. I think the discussion of the proactive suppression in the discussion might benefit with contrast to what is expected with retroactive suppression.

We discuss the effect of proactive and retroactive suppression on p. 21

3. It might be helpful to add to the Procedure where the experimental code, data and analysis code may be accessed to the manuscript.

We added it in the author notes. 

TYPOGRAPHICAL ERRORS

pp. 3. “Recently, it was pointed out that a third categories”, should be “a third category”.

pp. 7. The reference is missing a comma, “Wang, van Driel, et al. 2019).

pp. 10. “did not respond or respond incorrectly” should be “or responded incorrectly”.

pp. 12. “The F-value of the second paragraph has an extra period.

pp 12. “Again, there was significantly different” should be “there was a significant difference between…”

pp. 12. “Paired-wise t-test” should be “Pairwise t-tests”

All fixed.

Reviewer #3

In general I thought this was an interesting set of data and should make a nice contribution to the literature. I had a number of suggestions as to how I thought the authors could improve the manuscript. Points 1-3 I see as fairly critical, but I had a number of more minor points.

Thanks for your comments.

Point 1. The presentation of the rationale for the paper currently makes it seem like the contribution here is fairly modest. That is, similar effects have already been reported in the Wang and Theeuwes (2018) papers, and it wasn’t immediately clear what is left unanswered by those papers, and what the current paper addresses. I appreciate that the current paper provides a more direct test of suppression at various positions with the probe task, but I think that novel contribution, and its importance, can be made clearer in the introduction for those less familiar with the methods and results of previous papers.

Response: We make this clearer in the introduction (p. 7). Because we now present the full analysis of the probe task using both guess rate and standard deviation the contribution of this paper to the existing literature has also become clearer. Thanks!

Point 2. It isn’t clear exactly what the competing hypothesis is for this experiment. On page 6 the authors state that “Proactive suppression can be contrasted with retroactive suppression, which is the type of suppression that occurs only after attention has been directed to a location, disengaged and subsequently suppressed.” Could the authors expand upon this sentence and flesh out the range of possible results they considered and the theories they would support? I feel that this very brief treatment of the theoretical possibilities weakened the manuscript and it wasn’t clear how the current data sit with such alternative accounts. For example, consider the data shown in Figure 3B. If I’ve interpreted this correctly, these data suggest that in the absence of a distractor singleton, there is no impairment in probe responding at the high probability location relative to the low probability location. Doesn’t that suggest that it is indeed the presence of the highly salient singleton that attracts attention, and it’s only when that occurs, that active suppression comes into play? That sounds similar to what you describe above for “retroactive suppression”, but I didn’t see that mentioned in the discussion.

Response: This is an interesting and clever observation. Strictly speaking when only considering the response error score for the distractor absent condition I think the reviewer’s interpretation is correct. On the other hand, Figure 2b shows that selection of the target is hampered when it appears at the high probability location which suggests that this location is supressed proactively. The result has been replicated already 10 times in the different experiments of these studies (Wang & Theeuwes, 2018a,b,c). This result can only be interpreted as evidence for pro-active suppression. Why this effect is not found in the response error score of the probe is hard to explain but likely due to a floor effect. On the basis of this overall pattern of results we stick to the original interpretation and argue for proactive suppression 

Point 3. Given that this is an effect of learning the statistical properties of the spatial locations, and given that each participants contributed 2520 trials, it seemed to me a missed opportunity not to analyse how these patterns develop over time. Could the authors break the data down into blocks of trials so we can see the development of the effect?

Response: Thanks. We have reported in the Ms, see p. 13. Similar to what we have observed in our previous studies, people learned to suppress the high-probability location very quickly. The results showed that the mean RTs for the high-probability location was smaller than that for low-probability location in the first block, p < .001, and in the following blocks, all ps < .003.

Point 4. The description of the contextual cuing literature on page 4 doesn’t seem very accurate. Firstly, the statement “…participants were faster in finding targets when they appeared in repeated configurations than in novel locations” is ambiguous, in that “novel locations” could refer to the target location. Better would be “…participants were faster to find targets when they appeared in repeated configurations than when they appeared in novel configurations”. Secondly, I don’t agree that this is related to Posner cueing tasks, at least not in the way that is stated in the next sentence. Contextual cuing is generally not about learning the probability of where targets appear per se. Rather, it is about learning the association between configuration and target. Typically target location probability is controlled across repeated and random configurations.

Response: We have changed the wording 

Point 5. Having said that, there is a small literature on probability cuing in contextual cuing of visual search, which starts with the paper by Jiang et al. (2013). This will be of interest to the authors and may be relevant for the discussion here.

Jiang, Y. V., Swallow, K. M., & Rosenbaum, G. M. (2013). Guidance of spatial attention by incidental learning and endogenous cuing. Journal of Experimental Psychology: Human Perception and Performance, 39, 285–297.

Response: Added 

Point 6. pg. 9 – it wasn’t initially clear to me from Figure 1 how the participants knew which probe was being queried in the task. It is mentioned in the text briefly, but I found the figure a bit confusing, given the final panel presumably shows a magnified/schematic version of the screen. I think this Figure needs some work to clarify how the task operated, perhaps showing both the probe recall prompt in one of the actual positions, and then also magnified to show the detail.

Response: Thanks. We have changed the figure.

Point 7. I appreciate that only a small percentage of trials were removed as outliers, but I would prefer to see the upper limit on RTs be set on a participant basis and related to the distribution of RTs for that participant (e.g., 2.5/3 SDs above the mean RT). Otherwise, this is set fairly arbitrarily and it begs the question of why this value was selected. Were the results in any way contingent upon this upper threshold value?

Response: We have done all the analysis again with using 2.5 SD as a criteria to delete the outliers, and found the results pattern did not change. The way we deleted the data is according to the distribution of the RTs; we deleted the data belonged to two tails of the overall distribution.

Point 8. I applaud the authors for making their materials and data available. I downloaded these and I was able to read in the data files, but I couldn’t make much sense of many of the variables in the data files. I would suggest including a document that lays out clearly what the variables are, what the levels of those variables are coded as, etc. It would be great if the authors also included their analysis scripts.

Response: Thanks for pointing out this. We have uploaded it.

---

## [Decision Letter · Decision Letter 1]

6 May 2020

PONE-D-19-35559R1

Proactively location-based suppression elicited by statistical learning

PLOS ONE

Dear Mr. Wang,

Thank you for submitting your manuscript to PLOS ONE. After careful consideration, we invite you to submit a revised version of the manuscript that addresses the last few minor points raised during the review process.

I have received additional assessments from two of the original reviewers. The third has indicated that they are happy for me to proceed but won't be able to produce a review in the near future for reasons that are understandable given the current difficult circumstances that we find ourselves in. The two reviewers are largely happy with your revisions, if you can address the few very minor points that Reviewer 2 has identified below, I'll endeavor to make a decision quickly without sending it out for further review.

We would appreciate receiving your revised manuscript by Jun 20 2020 11:59PM. To enhance the reproducibility of your results, we recommend that if applicable you deposit your laboratory protocols in protocols.io, where a protocol can be assigned its own identifier (DOI) such that it can be cited independently in the future. For instructions see: http://journals.plos.org/plosone/s/submission-guidelines#loc-laboratory-protocols

We look forward to receiving your revised manuscript.

Kind regards,

Evan James Livesey, Ph.D

Academic Editor

PLOS ONE

Reviewers' comments:

Reviewer's Responses to Questions

**Comments to the Author**

1. If the authors have adequately addressed your comments raised in a previous round of review and you feel that this manuscript is now acceptable for publication, you may indicate that here to bypass the “Comments to the Author” section, enter your conflict of interest statement in the “Confidential to Editor” section, and submit your "Accept" recommendation.

Reviewer #1: All comments have been addressed

Reviewer #2: All comments have been addressed

2. Is the manuscript technically sound, and do the data support the conclusions?

Reviewer #1: Yes

Reviewer #2: Yes

3. Has the statistical analysis been performed appropriately and rigorously? 

Reviewer #1: Yes

Reviewer #2: Yes

4. Have the authors made all data underlying the findings in their manuscript fully available?

Reviewer #1: Yes

Reviewer #2: Yes

5. Is the manuscript presented in an intelligible fashion and written in standard English?

Reviewer #1: Yes

Reviewer #2: Yes

6. Review Comments to the Author

Reviewer #1: The authors successfully addressed all my comments. Also, the authors included the new model fitting analysis and showed an interesting result. I think this manuscript will be a good addition to Plos One. I have no further comments.

Reviewer #2: The authors have made revisions that have improved the manuscript and addressed the concerns in my initial review. A new mixture model analysis showed precision for probed recall was best at the target location compared to the other locations. More guesses occur when the probe is at the high-probability distractor singleton location than the low-probability distractor location indicating that the high-probability location is being proactively suppressed. I believe the manuscript can be accepted following these very minor revisions:

Typo on p. 17: The performance was worse for distractor singletons that appeared at the high-probability location than at the low-probability location..

Typo on p. 19: The guess rate was higher for distractor singletons that appeared at the high-probability location..

Figure 3 can be improved by jittering the bars horizontally so that the 95% CI are visible in each condition.

William Xiang Quan Ngiam

I sign all my reviews, regardless of the recommendation to the editor. By signing this review, I affirm that I have made my best attempt to be polite and respectful while providing criticism and feedback that is hopefully helpful and reasonable.

7. PLOS authors have the option to publish the peer review history of their article (what does this mean?). If published, this will include your full peer review and any attached files.

Reviewer #1: No

Reviewer #2: Yes: William Xiang Quan Ngiam

---

## [Author Response · Author response to Decision Letter 1]

7 May 2020

Reviewer #2

The authors have made revisions that have improved the manuscript and addressed the concerns in my initial review. A new mixture model analysis showed precision for probed recall was best at the target location compared to the other locations. More guesses occur when the probe is at the high-probability distractor singleton location than the low-probability distractor location indicating that the high-probability location is being proactively suppressed. I believe the manuscript can be accepted following these very minor revisions:

Typo on p. 17: The performance was worse for distractor singletons that appeared at the high-probability location than at the low-probability location.

Fixed.

Typo on p. 19: The guess rate was higher for distractor singletons that appeared at the high-probability location.

Fixed.

Figure 3 can be improved by jittering the bars horizontally so that the 95% CI are visible in each condition.

It is kind of weird to show a guess rate below 0, which is unusual in the memory literature. Moreover, when comparing two conditions, people would like to check the overlap between the error bars of different conditions, which could be detected from the current version of the figure. So, we chose to keep it.

---

## [Editor Report · Decision Letter 2]

8 May 2020

Proactively location-based suppression elicited by statistical learning

PONE-D-19-35559R2

Dear Dr. Wang,

We are pleased to inform you that your manuscript has been judged scientifically suitable for publication and will be formally accepted for publication once it complies with all outstanding technical requirements.

With kind regards,

Evan James Livesey, Ph.D

Academic Editor

PLOS ONE
---

## [Editor Report · Acceptance letter]

12 May 2020

PONE-D-19-35559R2 

Proactively location-based suppression elicited by statistical learning 

Dear Dr. Wang:

I am pleased to inform you that your manuscript has been deemed suitable for publication in PLOS ONE. Congratulations! Your manuscript is now with our production department. 

With kind regards,

on behalf of

Dr. Evan James Livesey 

Academic Editor

PLOS ONE